# Stability of Postcritical Deformation of CFRP under Static ±45° Tension with Vibrations

**DOI:** 10.3390/polym14214502

**Published:** 2022-10-25

**Authors:** Valeriy Wildemann, Oleg Staroverov, Elena Strungar, Ekaterina Lunegova, Artur Mugatarov

**Affiliations:** Center of Experimental Mechanics, Perm National Research Polytechnic University, 614990 Perm, Russia

**Keywords:** postcritical deformation, composite, digital image correlation, acoustic emission, stability

## Abstract

The paper presents an experimental study on regularities of postcritical deformation of carbon-fiber-reinforced plastic (CFRP) under static ±45° tension. The employed test method is based on ASTM D3518. Displacement and strain fields were identified by a digital image correlation method (DIC) using a VIC-3D contactless optical video system. Acoustic emission signals were obtained using an AMSY-6 system. The surface analysis of samples was carried out using a CarlZeiss SteREO Discovery. V12 optical stereomicroscope and a DinoLite microscope. Three experimental test types were considered: active loading, deformation with unloadings, and tension under additional torsion vibrations with various amplitudes. Loading diagrams were constructed; they showed a number of stages in the damage accumulation process. It was analyzed how heterogeneous strain fields develop; a neck development during softening process was observed. It was noted that the loading system rigidity influences the failure moment. The research considered various shear strain calculation methods using a “virtual extensometer” instrument. Composite mechanical properties were obtained. A shear modulus reduction during a plastic strain increase was revealed. The acoustic emission signals were analyzed; three characteristic frequency bands were observed. Most of the contribution to cumulative energy was made by matrix cracking. A reduction of the number of AE signals associated with the violation of adhesion between the fibers and the matrix during postcritical deformation was observed. The research identified basic surface defects. An appearance of the defects corresponds with their identification by the AE system. It was revealed that the presence of additional torsion vibration leads to an increase in the softening stage length. It was concluded that due regard for the postcritical deformation stage and the loading system rigidity is reasonable during the structure strength analysis.

## 1. Introduction

A prediction of destruction processes and an assessment of a structure’s survivability and safety require studying the issues of structural damage accumulation during deformation. These processes lead to the appearance of a nonlinear relationship between stresses and strains at the macro level. If structural defects are well-developed, the realization of the postcritical stage is possible. It is characterized by a stress decrease during increasing strains. Various areas of this stage correspond to the various construction destruction stages. Experimental and theoretical studies of the postcritical deformation stage were carried out in the works [1,2,3,4,5,6].

An important feature of this stage is the possibility of stability loss at various points. For example, under “soft” loading conditions with a permanently increasing force, destruction will occur when the stresses reach ultimate strength. On the other hand, in the case of “hard” kinematic loading, the implementation of the complete stress–strain curve is possible. Generally, the stability loss moment depends on the stiffness of the loading system (bodies that transmit the load), as was shown in the papers [6,7,8,9,10]. A number of works [11,12,13,14] demonstrated the effect of the increasing stability of the softening process in the additional vibrations presence.

Various materials undergo postcritical deformation: metals [15,16,17], rocks [18,19], concretes [20,21,22], polymer composites [23,24,25], etc. In addition, the softening effect is observed in the in-plane shear response of polymer composites [26,27,28,29]. However, this stage has not been studied in detail. We can conclude that postcritical stage experimental investigation is relevant.

Compared to traditional metals and alloys, CFRP have equal or higher physical and mechanical properties. CFPR are introduced into high-loaded constructions of the aviation, rocket, and automotive industries, where the issues of weight, strength, durability, functionality, and cost are relevant [30,31].

Researchers give preference to the non-contact optical video system VIC-3D during performing shear tests in accordance with various ASTM [32,33]. There is a research [34] that presents static tests with the results of unloadings for thermoplastic composite samples prepared in accordance with ASTM D3039. Registration of deformations on the samples surface was carried out using the GOM-Aramis DIC system. Another research [35] carried out an analysis of CFRP behavior under shear by various methods (ASTM D3518, ASTM D5379, and ASTM D7078) using a video system. Strungar et al. [36] presented experimental studies of the 3D reinforced composites’ shear properties in accordance with ASTM D5379. It was established that the video system usage gives an advantage over strain gauge because it allows one to see the development of the inhomogeneous strains.

In addition, a convenient method for incipient and developing structural damages identification is the acoustic emission (AE) method based on the registration of elastic waves that arise during the deformation and the local structure rearrangement [37,38]. The advantage of the AE method is the tracking possibility for internal damage and effective information provision for understanding the failure process. Furthermore, it is necessary to identify the composites structural destruction mechanisms. The researchers revealed the characteristic frequency ranges of the AE signals and their correspondence to the various damage accumulation mechanisms in the polymer composites (a matrix cracking: the lowest frequencies; fiber breaks: the highest frequencies; an adhesion failure between the fiber and the matrix: medium range) [39,40,41,42,43,44,45,46,47]. It should be noted that a limited number of works address the study of the composites postcritical deformation stage with the AE method.

Based on the above review, we consider that the study of the postcritical deformation regularities of polymer composites under in-plane shear is relevant. In particular, it is necessary to study the softening process stability and the influence of additional vibrations.

## 2. Material and Methods

### 2.1. Material

The flat samples with layout [±45]_16_ made on the basis of VKU-60 prepreg and VSE 1212 polymer matrix. The samples were cut from a single plate made by autoclave molding at a temperature of 175 °C and a pressure of 500 kPa. After molding, the plate was kept at 120 °C for 6 h to eliminate residual stresses and post-polymerization effects. The samples corresponded to the ASTM D3518 with the working area size of 200 mm× 25 mm.

### 2.2. Equipment

Experimental studies were carried out using the large-scale research facilities “Complex of testing and diagnostic equipment for studying properties of structural and functional materials under complex thermomechanical loading” at the Center of Experimental Mechanics of the Perm National Research Polytechnic University (PNRPU).

Tensile tests were carried out using the Instron 8802 (±100 kN)—a universal servohydraulic testing system with a movable grip speed of 2 mm/min. The loading was recorded by a local cell up to 100 kN. The loading measurement accuracy is 0.5%. The testing system includes a FastTrack controller with a feedback function, which allows a researcher to compensate the sample stiffness loss during the test. The WaveMatrix software allows implementing complex loading modes. The tensile test was set in the software in a stepwise manner.

Displacements and strains of the specimen surface were recorded using the VIC-3D contactless optical video system and the digital image correlation (DIC) method. A video recording of the deformation process was carried out using camera Prosilica 50 mm. The recording frequency was 1 frame for every 0.05 kN load change or 1 frame per 10 s with the maximum frequency of 5 Hz. The camera resolution was 16 MP. The use of the video system required the synchronization of the experimental data with the loading process. The software of the VIC-3D video system provides the usage of various correlation criteria for the mathematical assessment of the digital image correspondence. The normalized sum of squared difference criterion (NSSD) was used. This criterion is less sensitive to the sample illumination (brightness) changes during deformation and provides the best combination of the time frames and the results accuracy. During postprocessing, the VIC-3D system was used to calculate the strains using the Lagrange finite strain tensor.

The acoustic emission signals were recorded using the AMSY-6 system (Vallen Systeme GmbH, Wolfratshausen, Germany). We employed one wideband sensor AE144A (Fujicera, Fujinomiya, Japan) with a frequency range of 100–500 kHz and a preamplifier with a gain of 34 dB. The sensor was attached to the sample using a rubber fixture. The data sampling frequency was 10 MHz, and the threshold value for recording AE signals was 40 dB. The following parameters were considered informative: the energy parameter, the duration of the AE signals, and the frequency of the spectral maximum (characteristic of the fast Fourier transform). The energy parameter of AE signals was calculated using a special software option in energy units (eu), 1 eu = 10^−14^ V^2^∙s. To record the applied load and displacement, the AMSY-6 and the video system were synchronized with the test system controller using a 16-bit high-speed NI USB-6251 ADC unit.

After testing, the researchers used a CarlZeiss SteREO Discovery. V12 stereomicroscope with ZEN software in order to determine the sample surface defects. We employed a DinoLite microscope with 10× magnification mounted on a tripod and DinoCapture 2.0 software with the aim to track the surface changes during the tests. The photos of the diagnostic systems and the sample in the testing machine grips are shown in Figure 1.

### 2.3. Methods

Ten CFRP samples were tested; they were divided into three groups. In the first group (two samples), tensile tests were carried out until the samples failure (or until the load reduction to 0.5 kN) with the movable traverse speed of 2 mm/min. VIC-3D and AMSY-6 were used as additional diagnostic systems.

In the second group (four samples), loading was performed with a number of unloadings. The testing process can be represented as two steps: the first is the active loading up to the determined by an operator point; the second is the transition to the unloading down to the threshold load value of 0.25 kN, with the automatic transition to step 1. Tests were also carried out using a video system and an acoustic emission signal recording system. In two cases, a DinoLite microscope was used.

In the third group (four samples), the sample was actively loaded to failure with the involvement of additional torsion vibrations. For three samples, the rotation angle amplitudes were 0.25, 0.5, and 1 degrees, respectively, and vibrations were applied at a frequency of 20 Hz at the maximum-load-reaching moment. For one sample, vibrations with an amplitude of 0.5 degrees were applied from the start of the test. The experimental research program is presented in Table 1.

## 3. Results and Discussion

### 3.1. Active Loading and Postcritical Deformation Regularities

#### 3.1.1. Loading Curves

Loading curves (Figure 2) were built using the load and the traverse displacement data. We can see the postcritical deformation stage at the macro level (load decrease with displacement growth). Nonequilibrium load decline areas are shown as dashed lines. One unloading on sample 2 was performed with the displacement ≈16 mm; an elastoplastic material behavior was revealed. Curves may be conditionally divided into six stages: I—elastic deformation; II—almost linear hardening up to the maximum load value; III—initial postcritical deformation with the neck occurring (fast stage); IV—stabilization of postcritical deformation with neck development (slow stage); V—intensive softening, which leads to the nonequilibrium damage accumulation and is able to cause full damage (fast stage); VI—final sample separation with slow load decrease (fibers are drawn from the matrix). We can assume that there is a correlation between the loading stages and the various damage accumulation mechanisms.

#### 3.1.2. Shear Strain Calculation

According to the recommendations of ASTM D3518, the shear strains determination requires usage of two strain gauges installed in the sample center in directions parallel and perpendicular to the sample axis to measure longitudinal and transverse strain. Shear strain is defined as the difference between longitudinal and transverse strain. The VIC-3D software allows simulating strain gauges with the use of “virtual extensometer” tool. In addition, it is possible to switch to a coordinate system rotated by 45° associated with the reinforcement axes and to directly obtain the shear strain field. It is assumed that the use of the “rectangular area” tool is efficient for this purpose. Due to the neck formation on the sample, there occurs a significant localization of strains. The application of the video system facilitates the data averaging in a given area, which provides the more accurate strain measurement.

Various strain analysis tools are shown in Figure 3a on the right. A virtual pair of extensometers E_0_E_1_ was installed at the point where the sample was fractured (at the neck center). Additionally, the “rectangular area” tool R was placed in this area to average the shear strain field in the rotated coordinate system. Pairs of virtual extensometers were also installed over the entire surface of the sample both closer and further from the neck (E_2_E_3_, E_4_E_5_, and E_6_E_7_). For comparison, material stress–strain curves were built, shown in Figure 3a on the left. It is noted that the material located at a distance from the localization place practically did not pass to the postcritical stage. The use of the “rectangular area” tool allows getting the softening stage but only to a certain extent. This happened due to the presence of weakly deformed zones under separation area. The application of the pair of E_0_E_1_ extensometers allows obtaining an extended softening stage. Due to this, further calculations will be carried out following a similar method. The disadvantage of this method is the partial loss of data associated with the delamination of the paint applied to the surface.

Figure 3b represents the fields of transverse strains ε_xx_, longitudinal strains ε_yy_, shear strains ε_xy_, and shear strains ε^rot^_xy_ in the rotated coordinate system for point 4 on the stress–strain curve. The significant inhomogeneity of the strain field in the narrowing area is noted. The shear strain fields ε^rot^_xy_ for points 1–4 on the stress–strain curve are shown in Figure 3c. It is evident there is a significant heterogeneity and localization of strains in the form of strips located along the composite reinforcement lines. In addition, we note that there are individual spots of strain localization; this can be due to the CFRP weaving structure influence and a certain characteristic size of a periodicity cell.

#### 3.1.3. True Stress–Strain Curve

Some researchers assume that the postcritical stage is associated with a cross-sectional area change, leading to the load decrease. To confirm the presence of a softening stage, let us construct a stress–strain curve in true stresses τ_true_. Instead of dividing the load *F* by the doubled initial cross-sectional area *S*_0_, we divide it by the doubled current value of the cross-sectional area *S_current_*, which was calculated using the transverse strain ε_xx_ according to the formula
(1)$$\tau_{true}=\frac{F}{2S_{current}}=\frac{F}{2\left(1+\varepsilon_{xx}\right)S_{0}}.$$

It was assumed that the thickness of the sample changed slightly. Figure 4 shows stress–strain curves constructed with the use of engineering and true stresses. A significant difference in stresses is noted; however, the postcritical stage of deformation was observed in both cases.

#### 3.1.4. Acoustic Emission Signals Analysis

Figure 5a shows that the energy parameter of the acoustic emission signals is dependent (for each second) on the displacement; the graphs are combined with the loading diagrams. The first sharp increase of the energy parameter value occurred at the load value ≈6 kN. This moment corresponds to the transition from the elastic deformation (stage I) to the linear hardening (stage II). Subsequently, the energy parameter peaks were recorded at the load of 8 to 10 kN without visible changes in the loading diagram. During the postcritical deformation, the energy parameter value increases at fast stages III and V; the energy parameter maximum value corresponds with the sample macro-destruction moment.

Figure 5b shows the distribution diagram of the acoustic emission signal frequencies versus displacement. All the signals can be combined into three groups with frequency ranges 50–100 kHz, 190–240 kHz, and 250–350 kHz. According to the previous researches, we can assume correspondence of these ranges to the matrix cracking, delamination, and interfacial de-bonding and fibers breaking. Other frequency ranges were not considered due to the low number of signals.

The diagram of the cumulative energy value dependence on the displacement is shown in Figure 5c. The cumulative energy was obtained for all signals and for the three frequencies ranges introduced. Most of the contribution to cumulative energy was made by matrix cracking: up to ≈90% of the maximum load value. Then, the fiber breaks occurred. This effect can be explained by a reversal of the fibers in the composite. A cumulation energy value leap is related to the macro-destruction of the sample and can be explained by multiple fibers breaks.

It was noted that an acoustic emission signal had an increased duration effect (Figure 5d). The maximum duration values correspond to stages III and V of fast load decrease.

#### 3.1.5. Typical Defects after Failure

The analysis of defects occurring on the sample surface after failure was carried out using the optical stereomicroscope (Figure 6). Characteristic damages for the farthest from the neck sample parts are delaminations between the braiding tapes (1, 2), which occurred both near the edge and closer to the center. Moreover, interlayer delamination (6) is visible on the side surfaces. Violations of adhesion between fibers and matrix inside the tape (5) occurred closer to the neck. Breaks of fibers and tapes (3, 4) are visible at the neck center. The fibers at the neck center are nearly parallel to the sample axis. An explosive failure at the neck is visible. All the characteristic damages correspond to those detected with the use of AE technique.

### 3.2. Unloadings and Postcritical Deformation Stability

#### 3.2.1. Loading Diagrams and Stability Analysis

The loading diagrams are shown in Figure 7a. The dotted lines indicate unloading-reloading loops; the dashed lines show the nonequilibrium damage accumulation areas. For specimens 4–6, after unloading, postcritical deformation continued at displacements smaller than those at the start of unloading. Comparison between the loading diagrams of samples 1 and 4 (Figure 7b) shows the absence of nonequilibrium damage accumulation process. We see the so-called S-shaped loading diagram. This effect is explained by the influence of the sample parts located above and below the neck; they actually constitute a loading system with limited rigidity. To confirm this fact, loading diagrams for each of these parts were built. Virtual extensometers E_8_ and E_9_ were used, and their location is shown in Figure 7c on the right. Load diagrams are shown in Figure 7c on the left. Both these parts entered the softening stage, and then, they practically behaved like elastic elements with slightly changing plastic deformation during unloading-reloading. The damage accumulation processes occurred in the neck area. The postcritical deformation stability loss occurred at stage V after a sharp increase of the neck area stiffness.

#### 3.2.2. Mechanical Properties

The shear modulus and the shear strength were obtained using stress–strain curves, which were built using the method described in Section 3.1.2. The shear strength was determined as the stress at the strain value 0.05 (according to the ASTM D3518) and as maximum shear stress value. When the shear strain value reached 0.05, the test continuation allowed us to reveal the significant deformation and strength reserves (≈90%).

#### 3.2.3. Decrease of Mechanical Characteristics during Postcritical Deformation

It is reasonable to consider the plastic strain value influence on the shear modulus change and the shape of the unloading-reloading loop. The stress–strain curves obtained for specimens 4 and 6 are shown in Figure 8a. Six load-reload loops were received for both samples. For each loop, we calculated the plastic shear strain, shear modulus, unloading-reloading loop width, height, and their ratio. The results are presented in Table 2. Plastic deformation increase leads to the elastic modulus reduction; after reaching the ultimate strength, it decreases by more than three times. Figure 8b shows graphs of the shear modulus reduction as compared to the initial value; it occurs due to the plastic shear strain. We can also note a relative increase in the loop width with a shape change.

#### 3.2.4. Analysis of Acoustic Emission Signals

Figure 9a shows the dependence of the AE signal energy parameter on time. A small number of signals during the unloading-reloading loop was detected. They resembled Kaiser effect observed on rocks [48]. The maximum sharp corresponds to the multiple fiber breaks.

Distribution diagram of the AE signal frequencies versus time is shown in Figure 9b. We noted almost a complete absence of the 190–240 kHz range signals after the transition to stage IV. Due to the presence of a number of unloadings, the violation of adhesion between the fibers and the matrix is completed earlier if compared with active loading. The characteristic frequency ranges are similar to those described in Section 3.1.4.

Figure 9c shows dependences of AE cumulative energy on time. The cumulative energy started increasing in the third unloading-reloading loop for signals in the frequency range 50–100 kHz (similar to the sample subjected to active loading without unloading). Results are similar to those described in Section 3.1.4.

Increased duration effect was noted for AE signals (Figure 9d). Maximum duration values correspond to the III and V fast load decrease stages.

#### 3.2.5. Surface Defects Development during Tension

The investigation of the surface defects development during tension was carried out with the use of the optical microscope DinoLite. The loading diagram is shown in Figure 10a; the sample surface pictures for the points 1–12 are shown in Figure 10b. Only delaminations between the tapes occurred until the maximum load value (1–5). Then, as the neck developed (stages III and IV), violations between the fibers and matrix inside the tape were found (6–8). A significant reinforcement system rotation was noted. During the transition to the intensive softening (stage V), all connections between the tapes were broken in the strain localization area (9–10). After that, the dynamic fibers destruction followed, leading to sharp load decrease (10–11). Then, the final sample separation occurred with a slow load decrease (stage VI) and with fibers being drawn from the matrix. We noted that the appearance of the defects on the sample surface corresponds with their identification by the AE system.

### 3.3. Additional Vibrations Influence

#### 3.3.1. Loading Diagrams

Figure 11a shows the obtained loading diagrams (the dashed line shows the areas of nonequilibrium damage accumulation, accompanied by load decrease). Their comparison with the active loading diagram (Figure 11b) reveals a noticeable increase in the duration of neck development stage IV; a transition to an almost horizontal line was noted. However, the further dynamic sample destruction became more sudden, and stage VI was not realized. The torsional vibrations amplitude increase in the considered range, as well as their application from the start of the test, did not have a significant effect.

#### 3.3.2. Analysis of Acoustic Emission Signals

Dependence of the AE signal energy on displacement is shown in Figure 12a. A burst of AE energy was noted during the transition from elastic deformation to linear hardening. Results are similar to those described in Section 3.1.4.

Distribution diagram of the AE signal frequencies versus displacement is shown in Figure 12b. AE signals are also grouped into the three ranges described earlier in Section 3.1.4. and Section 3.2.4. There is an almost complete absence of AE signals in 190–240 kHz frequency range (corresponding to a decrease of adhesion between fiber and matrix) at the softening stage.

Dependences of the AE cumulative energy on displacement are shown in Figure 12c. In the presence of additional vibration effects, the share of energy released during the fibers destruction increased. We observed a linear increase in cumulative energy from the middle of stage II to the beginning of stage V.

Dependence of the AE signal duration on displacement is shown in Figure 12d. At the postcritical deformation stage, an AE signal duration increased by 2–3 times.

## 4. Conclusions

This work carried out an experimental study of the postcritical deformation regularities of the CFRP under in-plane shear. The test method is based on ASTM D3518. Three types of tests were considered: active loading, loading with unloadings, and loading with additional torsion vibrations.

The research revealed deformation process staging. The neck formation on the sample and the presence of plastic deformations were noted. The method for the shear strain determination was chosen with the use of the VIC-3D video system. The advantage of the video system employment in comparison with strain gauges was noted. The research revealed a significant localization of shear strains in the neck area. It was demonstrated that the softening stage retains even when plotting the stress–strain curve in true stresses. Acoustic emission signals were analyzed using the AMSY-6 system. The presence of three characteristic frequency ranges was noted. The main contribution to the cumulative energy was made by the matrix destruction. The analysis of the sample surface defects was carried out using optical microscopes. We noted the presence of delaminations between the composite tapes, interlayer delaminations, fiber breaks, and adhesion failures between the fibers and the matrix inside the tapes.

The stability of postcritical deformation was confirmed by carrying out a number of unloadings. We noted the influence of the loading system stiffness (sample parts above and below the neck). The material mechanical characteristics were calculated: significant deformation and strength reserves were revealed. The shear modulus decrease dependence on the plastic shear strain was demonstrated. A correspondence between the deformation stages and defects occurrence was determined with the use of optical microscope.

We have studied the influence of additional torsion vibrations on postcritical stage implementation. It was shown that vibrations led to the deformation resource increase for the composite. We observed a reduction of the number of signals associated with the violation of adhesion between the fibers and the matrix during postcritical deformation. This effect is especially pronounced in the presence of vibrations.

We conclude that it is rational and expedient to take into account the postcritical stage of deformation and the loading system stiffness during a structure’s strength analysis to identify additional strength and deformation reserves.

## Figures and Tables

**Figure 1 polymers-14-04502-f001:**
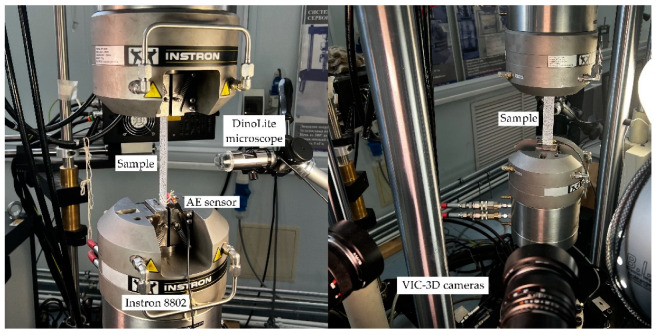
Sample in the testing machine grips and the diagnostic systems.

**Figure 2 polymers-14-04502-f002:**
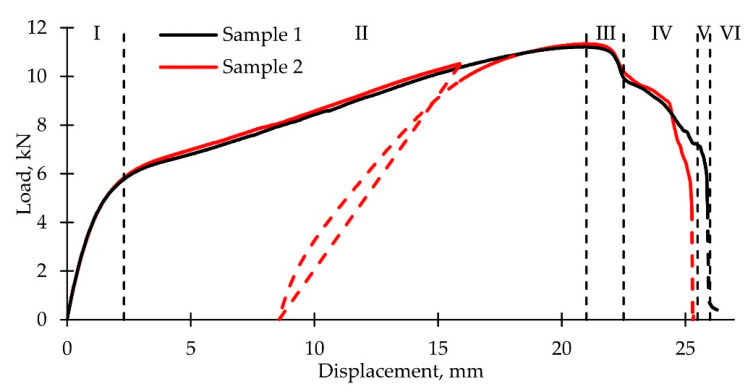
Loading diagrams with the noted stages: I, elastic; II, linear hardening up to maximum load; III, initial postcritical deformation with the neck occurring; IV, stabilization of postcritical deformation with neck development; V, intensive softening, which leads to the nonequilibrium damage accumulation; VI, final sample separation.

**Figure 3 polymers-14-04502-f003:**
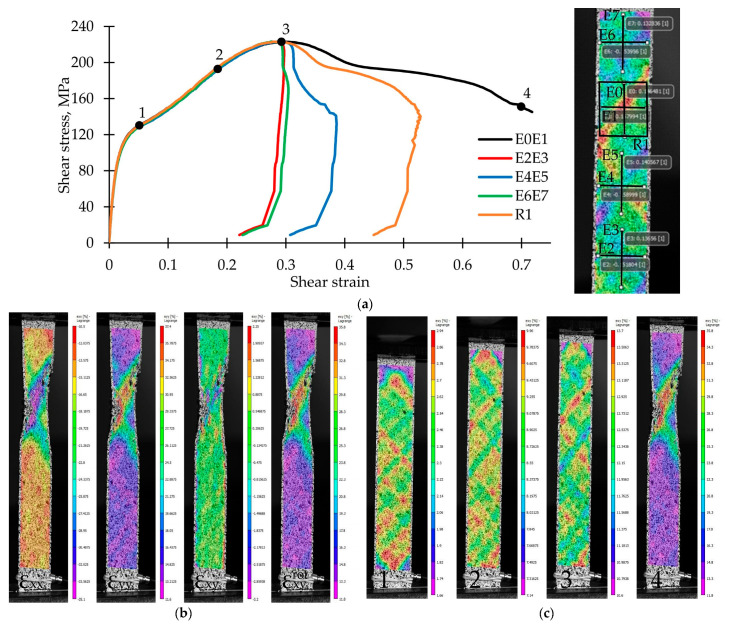
Choice of shear strain calculation method: (**a**) Stress–strain curves that are built following various methods (on the left); “virtual extensometers” location on the sample surface (on the right); (**b**) strains fields for point 4 on the stress–strain curve; (**c**) shear strains field evolution in the rotated coordinate system for the points 1–4 on the stress–strain curve.

**Figure 4 polymers-14-04502-f004:**
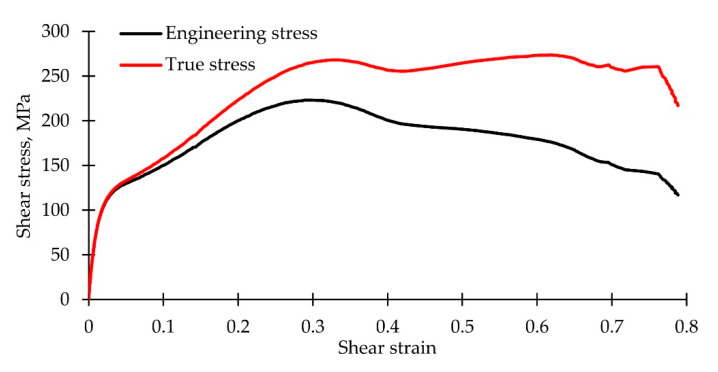
Stress–strain curves constructed with the use of engineering and true stresses.

**Figure 5 polymers-14-04502-f005:**
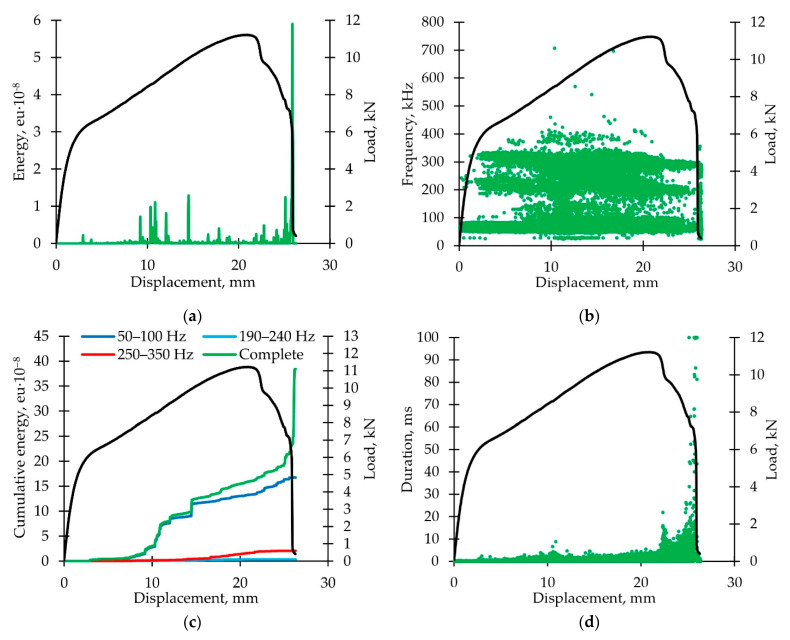
Analysis of acoustic emission signals (active loading): (**a**) Typical distribution diagram for the energy parameter of AE signals versus displacement; (**b**) typical distribution diagram of AE signals frequency parameter versus displacement; (**c**) typical diagram of cumulative energy dependence on displacement for various AE signals frequencies; (**d**) typical distribution diagram of AE signals durability parameter versus displacement. All the diagrams are accompanied by load diagram.

**Figure 6 polymers-14-04502-f006:**
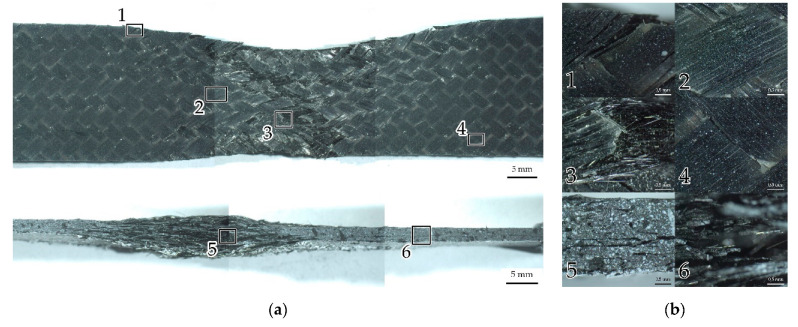
Sample after destruction: (**a**) Front and side surfaces close up; (**b**) magnified images of local areas. Numbers indicate typical damages: 1, 2—delamination between the tapes; 3, 4—breaks of fibers and tapes; 5—violation of adhesion between fibers and matrix inside the tape; 6—interlayer delamination.

**Figure 7 polymers-14-04502-f007:**
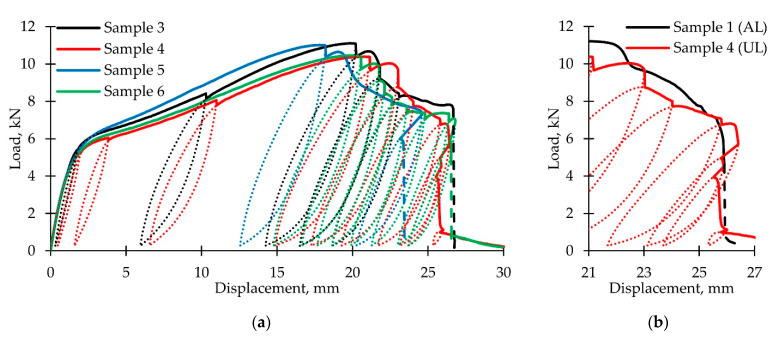
Loading-unloading curves: (**a**) Experimental data; (**b**) comparison between postcritical stage during active loading and loading-unloading; (**c**) loading diagrams of the sample parts above (E_8_) and below (E_9_) the neck.

**Figure 8 polymers-14-04502-f008:**
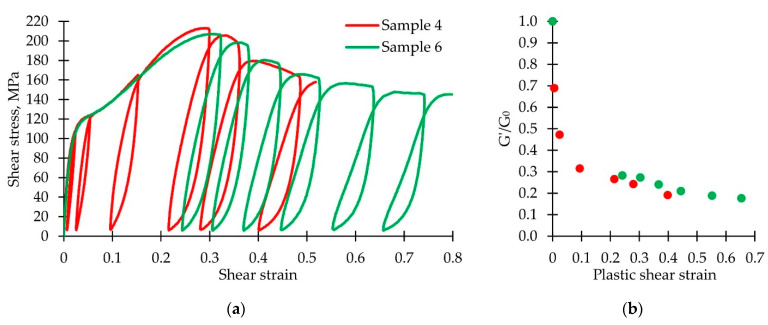
(**a**) Stress–strain curves; (**b**) shear modulus dependence on plastic strain.

**Figure 9 polymers-14-04502-f009:**
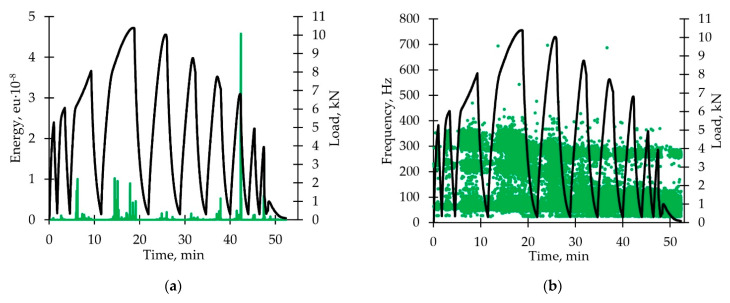
Acoustic emission signals analysis (loading with unloadings): (**a**) Typical distribution diagram of AE signal energy parameter versus time; (**b**) typical distribution diagram of AE signal frequency parameter versus time; (**c**) typical diagram of cumulative energy dependence on time for various AE signal frequencies; (**d**) typical distribution diagram of AE signal durability parameter versus time. All the diagrams are shown with load dependence on time.

**Figure 10 polymers-14-04502-f010:**
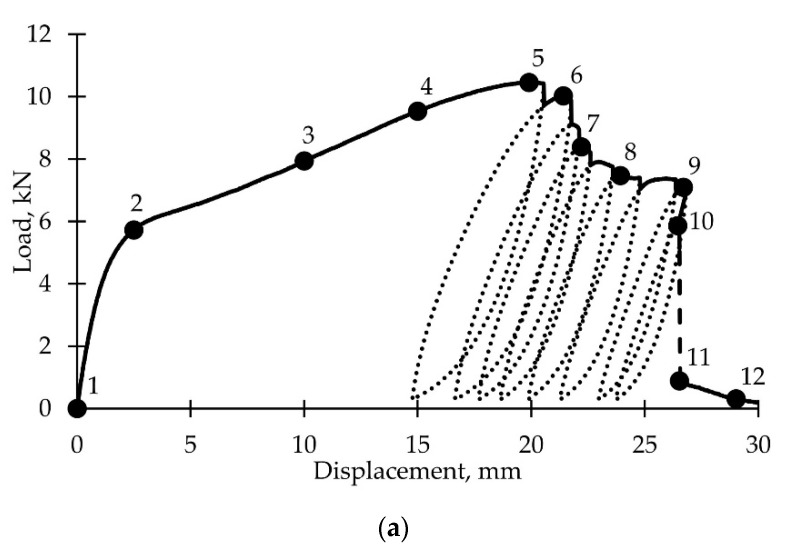
The surface defects development during the tension: (**a**) Loading diagram; (**b**) surface pictures corresponding to points 1–12 on the loading diagram.

**Figure 11 polymers-14-04502-f011:**
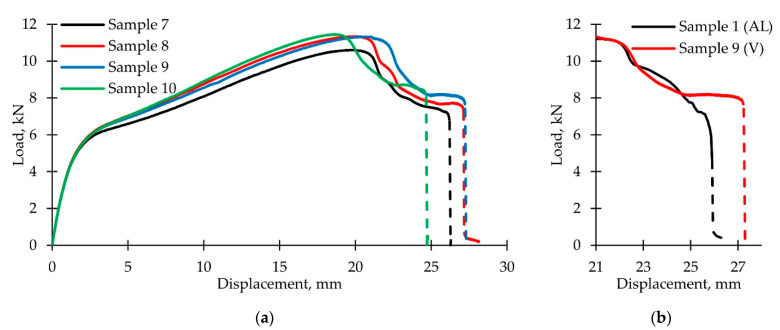
Loading diagrams with additional torsion vibrations: (**a**) Experimental data; (**b**) comparison between postcritical stage during active loading and loading with additional vibrations.

**Figure 12 polymers-14-04502-f012:**
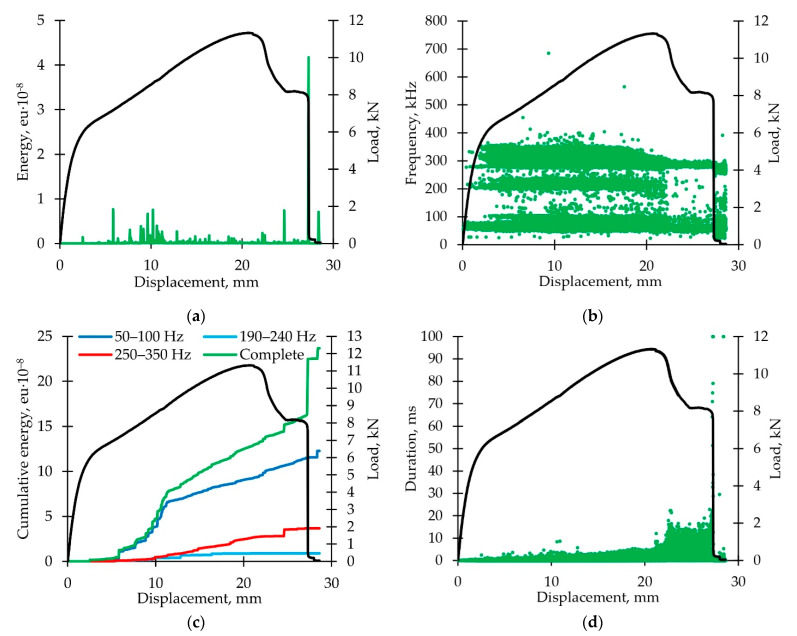
Analysis of acoustic emission signals (loading with additional torsion vibrations): (**a**) Typical distribution diagram of AE signal energy parameter versus displacement; (**b**) typical distribution diagram of AE signal frequency parameter versus displacement; (**c**) typical diagram of cumulative energy dependence on displacement for various AE signal frequencies; (**d**) typical distribution diagram of AE signals durability parameter versus displacement. All the diagrams are accompanied by load diagrams.

**Table 1 polymers-14-04502-t001:** Experimental program. AL, active loading; UL, loading with unloadings; V, loading with torsion vibrations.

Sample Number	Experiment Type	VIC-3D	AMSY-6	DinoLite
1	AL	+	+	–
2		+	+	–
3	UL	+	+	–
4		+	+	–
5		+	+	+
6		+	+	+
7	V	–	+	–
8		–	+	–
9		–	+	–
10		–	+	–

**Table 2 polymers-14-04502-t002:** Mechanical characteristics decrease and unloading-reloading loops analysis.

Sample Number	Loop Number	Plastic Shear Strain	Relative Shear Modulus G’/G_0_	Loop Width	Loop Height, MPa	Height/Width, MPa
4	1	0.0057	0.6893	0.0172	99.9	5808
	2	0.0242	0.4725	0.0286	114.1	3990
	3	0.0939	0.3154	0.0574	152.4	2655
	4	0.2138	0.2661	0.0831	186.8	2248
	5	0.2794	0.2427	0.0808	163.5	2024
	6	0.3983	0.1911	0.0851	136.9	1609
6	1	0.2410	0.2829	0.0788	178.0	2259
	2	0.3035	0.2728	0.0751	161.6	2152
	3	0.3673	0.2403	0.0761	144.2	1895
	4	0.4442	0.2103	0.0794	131.5	1656
	5	0.5517	0.1890	0.0843	122.4	1452
	6	0.6534	0.1765	0.0839	117.8	1404

## Data Availability

Not applicable.

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
