# Peer review of "Stability of Postcritical Deformation of CFRP under Static ±45° Tension with Vibrations"

_polymers, 2022, doi:10.3390/polym14214502_

Round 1

Reviewer 1 Report

The paper presents an experimental study of the postcritical deformation regularities of the CFRP under in-plane shear. Three types of tests were considered: active loading, loading with unloading, and loading with additional torsion vibrations.

The article is of interest to readers of the Journal. However, the authors should correct the manuscript and figures for clarity:

1. In page 2, line 79, area size should be corrected: 200 × 25 mm -> 200 × 25 mm2.

2. In page 3, lines 115,116, what is the stereomicroscope used for?

3. In page 8, Figure 6(a), rectangles and numbers hard to see.

Reviewer 2 Report

An interesting study will be conducted on stability of postcritical deformation of CFRP under static tension with vibrations. DIC and AE are used to obtain the deformation and defects of materials during the loading. The results are valuable for analyzing the internal damage of CFRP materials. The following comments should be considered for further improvement.

1. In the abstract part, the authors are encouraged to provide more results and conclusions. The introduction on the materials preparation and test/analysis methods should be further simplified.

2. This paper mainly focuses on the deformation stability of CFRP composites. However, the reviewers did not see any relevant introduction about this composite in the introduction. The performance, advantages and application should be summarized to provide the authors with a basic understanding for CFRP. In addition, the monitoring methods and performance evaluation of composite materials through DIC should also be supplemented with relevant research work. Please review the following latest research to fill this gap. Performance analysis: Construction and Building Materials, 2022, 315: 125710. Polymers, 2017, 9(11): 603. DIC analysis: https://doi.org/10.1080/15376494.2021.1974620.

3. Please provide more information about raw materials, for example, the composition, preparation method, manufacturer and other material parameters available.

4. The logic writing of 2.2 and 2.3 is relatively confused. It is recommended to adjust to the tensile testing and monitoring methods. In fact, equipment is also part of the test.

5. How do the acoustic emission signals analysis match the failure mode? Why not conduct some microscopic analysis and characterization, such as SEM? This can work well with acoustic emission to explain the defects, damage and failure.

6. In part 3.2.5, surface defects development during the tension, it is difficult to distinguish the defects through visual observation as shown in Figure 10.

7. In part 3.3, why do you study the additional vibrations influence? What is its application backgrounds? It is recommended to analyze this.

8. AE signals is considered to be a very effective means to detect the defects. However, it should also be used together with some microscopic analysis to confirm the formation, evolution, expansion of defects.

9. Conclusions, it is suggested to divide them into 3~4 points. according to the content of this paper.

Round 2

Reviewer 2 Report

Accepted.